# Performance, complexity and dynamics of force maintenance and modulation in young and older adults

Hester Knol[1,2], Raoul Huys[3], Jean-Jacques Temprado[1], Rita Sleimen-Malkoun[1]*

**1** Institut des Sciences du Mouvement, Centre National de la Recherche Scientifique (CNRS), Aix-Marseille Université, Marseille, France, **2** Department of Applied Cognitive Psychology, Universität Ulm, Ulm, Germany, **3** Centre de Recherche Cerveau & Cognition, UPS, CHU Purpan, Université de Toulouse, Toulouse, France

* rita.sleimen-malkoun@univ-amu.fr

**Data Availability Statement:** All relevant data are within the paper.

**Funding:** HK was funded through the Vieilstrat project granted by the French National Research

## Abstract

The present study addresses how task constraints and aging influence isometric force control. We used two tasks requiring either force maintenance (straight line target force) or force modulation (sine-wave target force) around different force levels and at different modulation frequencies. Force levels were defined relative the individual maximum voluntary contraction. A group of young adults (mean age ± SD = 25 ± 3.6 years) and a group of elderly (mean age = 77 ± 6.4 years) took part in the study. Age- and task-related effects were assessed through differences in: (i) force control accuracy, (ii) time-structure of force fluctuations, and (iii) the contribution of deterministic (predictable) and stochastic (noise-like) dynamic components to the expressed behavior. Performance-wise, the elderly showed a pervasive lower accuracy and higher variability than the young participants. The analysis of fluctuations showed that the elderly produced force signals that were less complex than those of the young adults during the maintenance task, but the reverse was observed in the modulation task. Behavioral complexity results suggest a reduced adaptability to task-constraints with advanced age. Regarding the dynamics, we found comparable generating mechanisms in both age groups for both tasks and in all conditions, namely a fixed-point for force maintenance and a limit-cycle for force modulation. However, aging increased the stochasticity (noise-driven fluctuations) of force fluctuations in the cyclic force modulation, which could be related to the increased complexity found in elderly for this same task. To our knowledge this is the first time that these different perspectives to motor control are used simultaneously to characterize force control capacities. Our findings show their complementarity in revealing distinct aspects of sensorimotor adaptation to task constraints and age-related declines. Although further research is still needed to identify the physiological underpinnings, the used task and methodology are shown to have both fundamental and clinical applications.

Agency (ANR-Programme Blanc SHS 2-2013). The funding agency had no role in study design, data collection and analysis, decision to publish, or preparation of the manuscript.

**Competing interests:** The authors have declared that no competing interests exist.

## Introduction

Many daily manual manipulation activities involve the maintenance and/or the modulation of a given force. In any case, it requires a continuous adjustment of the applied force to task constraints (e.g., duration, required precision, force level), as well as to the characteristics of the manipulated object, in order for instance not to squeeze or drop it. Such force tuning faculties are compromised by healthy aging, which is known to alter the overall possibilities and means of action of the organism (so-called intrinsic dynamics). Indeed, as we age, the properties of the neuro-musculo-skeletal system and the interactions between the different functional domains (e.g., cognition, motor control, cardiovascular capacities) are changed [1], with force control capacities being on the frontline with respect to sensorimotor impairments [2,3]. In the present work, we adopted a broad perspective to characterize force control in young and older adults using a functionally relevant task that can be quantitatively manipulated to scale the imposed attentional and sensorimotor requirements. We also combined conventional measures of performance, inferring precision and variability, with non-linear measures of signal complexity and dynamical components underlying force control. These two latter relate respectively to the adaptability and the generating mechanisms of behavior.

For a long time, age-related alterations of force control capacities have been attributed to peripheral changes in muscle strength and muscle structure or mass (so called sarcopenia, [4]). More recently however, it has been shown that they originate predominantly in the way the neuro-musculo-skeletal system recruits motor units in response to task requirements (i.e., in the central and peripheral nervous system, often referred as dynapenia [2,3,5,6]. In fact, aging compromises force control through the interplay of muscular [7–9], neural [2,3,6], cardiovascular [10,11], and cognitive alterations [12,13]. Such modifications in multiple sub-systems and their interactions are at the origin of the reduced behavioral capacities with advanced age. This reduction is often modulated by the characteristics of the considered task. For instance, force accuracy was found to be more altered by aging in tasks that require a low percentage of the maximum voluntary contraction (MVC) [14–17], and in tasks of a more challenging nature, like in sinusoidal force modulation [3,18]. These findings put forward a key functional aspect, which is adaptability. It corresponds to the capacities of the organism to produce adequate and task-specific responses that are both stable and flexible.

The complexity of physiological and behavioral signals is a macroscopic marker of health and adaptability in many functional domains (e.g., cardiovascular, metabolic and pulmonary functioning, mental health, gait, and force control) [1,10,11,19,20,21,22,23]. A complex signal can be defined through the presence of highly irregular, yet somewhat structured, fluctuations (long-range correlations). Signals' complexity is considered to reflect the connectivity between and within the different functional domains. It is a proxy of the richness of the repertoire of behavioral possibilities, which in return determines the adaptability of the organism to the environment and task constraints [1,20,24]. Aging has been shown to affect the structure of force fluctuations in force control tasks through mainly entropy-based metrics [18,25], with the direction of change (reduced or increased regularity) depending on the nature of the task. Specifically, older adults were shown to exhibit a more regular behavior than young adults when producing a constant force, and the reverse (less regular) when achieving a force modulation task [18]. However, in order to determine whether aging leads to changes in complexity, and not only regularity, we used here the multiscale entropy metric (MSE, [26]). Such a measure presents the advantage of differentiating between complex signals (i.e., irregular but with multiscale auto-correlation structure conferring high entropy over most timescales, e.g., pink noise) and random signals (i.e., completely irregular with a fast decay of entropy over timescales, e.g., white noise) [26,27].

From the dynamics perspective, biological systems are explicitly considered to inherently follow lawful principles (i.e., are deterministic), while being subject to random fluctuations on various scales rendering their dynamics stochastic. Both the deterministic (related to the strength of the behavioral attractor) and the stochastic (noise-like) components of the dynamics are known to contribute to the variability of the produced force over the time [28]. Indeed, in a constant isometric force production task, Frank et al. (2006) showed that with increasing force level (%MVC), the fluctuations of the produced force increase due to both a reduction in the stability of the fixed point (attractor) and an increase in the stochastic impact thereof. Only one subsequent study [3] explored the dynamical signatures with respect to task, age (i.e., young and middle aged), and expertise, in both a constant [28] and sinusoidal force modulation task. The stochastic and deterministic components were found to structurally differ between the two tasks regardless of age and expertise, suggesting that force maintenance and force modulation tasks have distinct dynamical signatures. On the one hand, the deterministic dynamics identified under constant force production was clearly associated with a linear fixed point [28]. On the other hand, for the sinusoidal force production, the structure of the deterministic component could not be unambiguously identified, but resembled a periodic orbit (i.e., a limit cycle dynamics). With respect to age-group differences, they were found to be maximized in the modulation task compared to the constant one, for both the deterministic and the stochastic components. The observed interaction between age- and task-related changes add to the evidence that skillful movements result from the interplay of internal (individual-related) and external (task-related) constraints [29]. Accordingly, changes in behavioral dynamics (i.e., a description of the components contributing to the variability of the overt behavior) reflect how well the underlying intrinsic dynamics (i.e., the possibilities and means of action of the neuro-musculo-skeletal system) is adaptable to task demands, and can therefore be related to the functional status of the organism [1,24]. Here, we sought to extend the available knowledge on the dynamic processes involved in task- and age-related adaptations in force control by including richer task conditions (two force levels and two modulation frequency) and an older adult group (compared to just young in [28]; and young and late-middle aged groups, in [3]).

Overall, we aimed at characterizing force control capabilities and properties in young and older adults when accomplishing isometric force control tasks with different requirements. Therefore, we analyzed performance in terms of precision (mean force, variance, coefficient of variation and root mean squared error), as well as the complexity (MSE) and the dynamics (stochastic and deterministic components) of the produced behavior. In that regard, while the precision variables are of clinical relevance, and allow for comparison with other force production studies, they contain little to no information as to the underlying force control system's functional organization. In contrast, the complexity measure provides informs about the time scales involved in the processes underlying force production, whereas the analysis of the dynamics separates the deterministic, and thus lawful, functional component of the performance dynamics from stochastic fluctuations. These analyses are thus complementary in characterizing force production. Increasing force level was expected to scale the statistical properties and the magnitude of force signals fluctuations [25,30] without altering the generating mechanism of the dynamics. We also expected to find the previously shown age × task interaction effect on the structure of force fluctuations. We hypothesize that this interaction should result mainly from the reduced functional range of behavioral complexity reflecting an impoverishment of the intrinsic dynamics of the aging neurobehavioral system [1], which precludes a further adaptation to accommodate the sinusoidal target as compared to the constant one. This finding could be related to a noisier (diffusion component) or less stable (drift component) dynamics in the elderly. More specifically, for the dynamics, although we expected to

identify a fixed point generating mechanism in the constant force condition versus a limit cycle in the sinusoidal one, we hypothesized that in the latter, decreasing the frequency could lead to a transition from a smooth oscillatory behavior to a bi-stable fixed point behavior [3].

## Materials and methods

### Participants

Thirty healthy adults voluntarily participated in the experiment after having given their informed written consent. All participants reported to be right-handed, having normal or corrected-to-normal vision, not having any neurologic or cognitive disorder including memory loss, not suffering from any musculo-skeletal disorder or pain that could compromise their performance in the task. Healthy cognitive functioning was assured using the Montreal Cognitive Assessment with the originally proposed cutoff score of 26 (MoCA; [31]). One participant was excluded from further analysis due to a low MoCA score. The protocol agreed with the Declaration of Helsinki and was part of a research program that was approved of by the regional ethics committee for biomedical research (CPP SUD MÉDITERRANÉE I, n° 2014-A01292-45).

Based on their age, the participants were divided into two groups: young participants (YN: 18; mean age ± SD = 24.9 ± 3.56 years; 6 females), and elderly participants (EL: 12; mean age ± SD = 76.69 ± 6.41 years; 8 females). Our participants produced isometric forces in a pinch grip with feedback given on a computer screen. In the following we detail the used setup.

### Experimental setup

A height-adjustable force transducer (SCAIME, ZFA, 50kg) was affixed to the experimental table, so that the participant could comfortably grasp it while being seated with their arms resting on the table (Fig 1). Participants were instructed to apply forces on the force transducer using a pinch grip with their right (dominant) index finger and thumb, while the other fingers built a fist. The produced force signal was recorded with a sampling rate of 240 Hz. A customized LabVIEW (National Instruments, Austin, TX) program was used to collect the produced force data and to provide on-screen visual feedback sampled at 12 Hz. The target force level and the actual grip force produced by the participants were displayed in red and yellow, respectively, over a black background on a 19-inch Dell Monitor with a 60Hz frame rate (Fig 1). The monitor was placed at a 60 cm viewing distance from the participant. The visual gain was kept constant and proportional to the MVC, with 1% MVC of applied force corresponding to an on-screen displacement of 0.31 cm.

### Task and procedure

As task conditions were defined relative each participant's force production capacity, we firstly measured the MVC of the right-hand index-thumb pinch. It was determined in three maximum pinch grip trials of 4 s by averaging force values recorded in the 2–3 s intervals. The same procedure was repeated following the completion of the task.

In the experimental task, participants were asked to match their produced force with an on-screen displayed target line as precisely as possible. The target force level and the produced pinch force in the time moved from the left to the right on the screen. The target curve was either a straight line or a sine wave (Fig 1). In both conditions, two mean force levels were used: 10% and 30% MVC. In addition, the sine wave was set at either a frequency of 0.4 or 0.8 Hz, with an amplitude of 8% MVC. The participants performed two series of 3 trials per

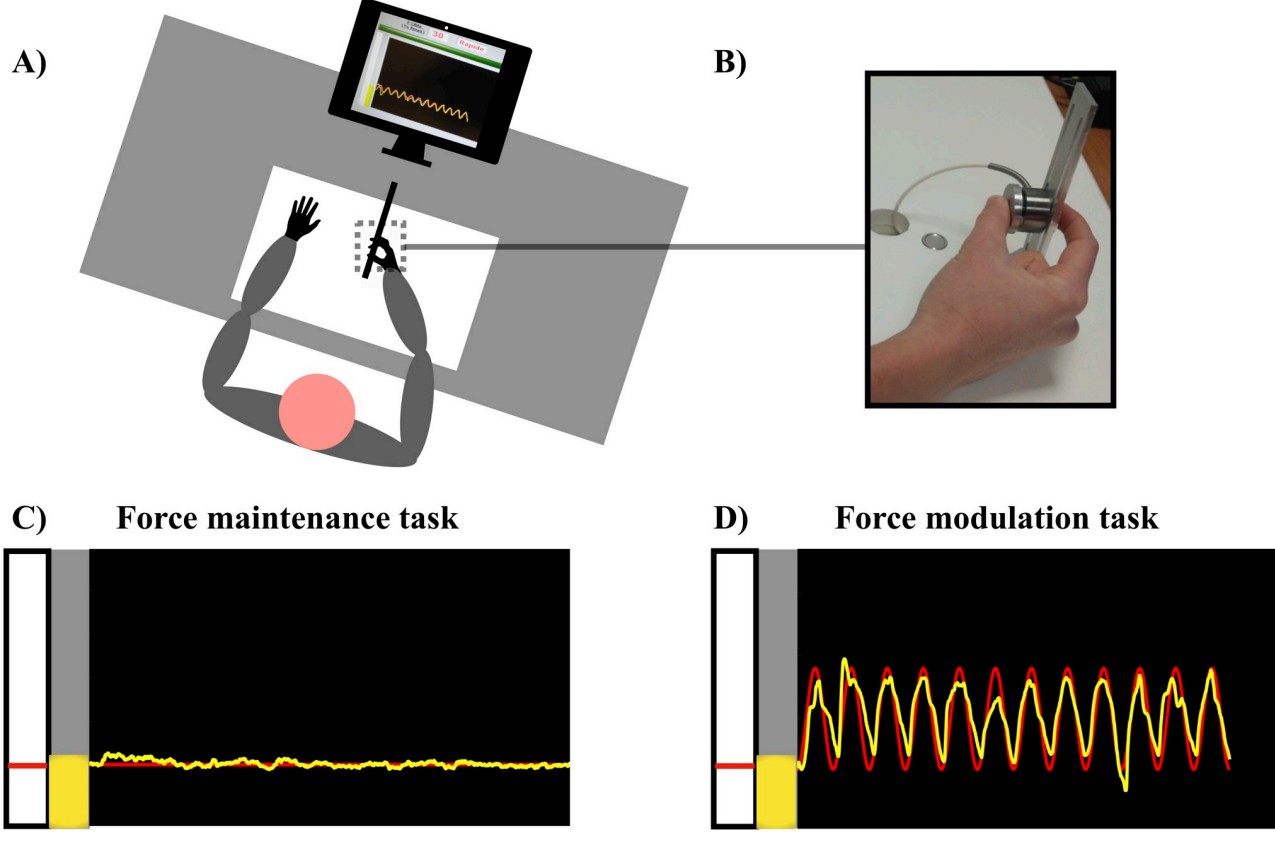

**Fig 1. Experimental set-up.** (A) Illustration of the experimental set-up showing the position of the participant facing the computer screen where the visual feedback is given, and holding the height-adjustable force transducer. (B) Close-up on a young participant holding the force transducer in a pinch-grip with their dominant hand. (C) Screenshot of the visual feedback during an exemplary constant force trial performance. (D) Screenshot during an exemplary sinusoidal force trial performance. The participants were required to squeeze the transducer (B) in order to move-up the on-screen yellow bar over the grey background, and match the red target line that appeared in the white space. Then, the red line would draw from left to right either as (C) a straight line or (D) a sine wave that the participant should keep matching with the yellow curve driven by their produced force.

condition in a randomized order, with a total of 36 trials. The order of the sine wave and constant force condition was counter-balanced among participants. The trials lasted 30 s for the 0.4 Hz and 15 s for the 0.8 Hz sine wave condition (12 cycles each), and 22 s for the constant force condition. The trial acquisition started after the participants had matched their produced force to the starting force of the trial (i.e., 10% or 30% MVC). The produced force was indicated by a vertical bar, and a red line on this bar indicated the starting force. Between the trials, participants had time to rest. To make sure the participants understood the task, the experiment started with familiarization trials with a minimum of 1 trial in each force production modality (constant and sine wave with a total of 6 trials).

## Data analysis

Data were analyzed using Matlab R2016a (MathWorks, Natick, MA, USA). Data were filtered with a 4th order low pass no-phase lag Butterworth filter with a cut-off frequency of 30 Hz.

For the force maintenance task, outliers were identified as exceeding the mean variability by ±2.5 times the standard deviation (SD), calculated per participant per condition (3 for a similar procedure [29]) or a force dropping below 0.1 N. The shortest part of the time series until, or between the outliers was excluded from further analysis. For the sinusoidal force task,

outliers were identified as either cycles in trials in which the force dropped below 0.05 N, or trials in which the amplitude exceeded 2.5 times the SD of the target. If the amplitude in the first period did not exceed 1.5 N, this period was excluded from further analyses.

**General characteristics of task performance.** The mean of the produced force, based on real force values, and the SD of the difference between the applied and the target force were calculated to globally capture accuracy and the amount of variability of force production.

The task performance was further quantified by the root mean squared error (RMSE) between the produced force and the on-screen target (Eq 1),

$$RMSE = \sqrt{\frac{1}{N-1}\sum\nolimits_{i=1}^{N}(x_i - T)^2} \qquad (1)$$

where $N$ is the number of data points in the time series, T is the force target, and $x_i$ is the force sample.

For the sinusoidal force modulation task, we computed the Hilbert phase of the applied force and the target for each trial. The relative phase, i.e., the difference between the applied force phase and the target force phase, was positive when the applied force lagged the target. We next calculated the mean and variance of the relative phase using circular statistics [32]. These measures provide information about the accuracy and the variability of the relation between the target curve and the applied force.

**Multiscale entropy.** The variability structure of the force time series was characterized using multiscale entropy (MSE) with a modified coarse-graining procedure yielding smoother curves [33].

MSE consists of calculating the sample entropy (SampEn; [34]) over multiple time-scales contained in the analyzed time series, and thus in the expressed dynamics [35,36]. SampEn measures the degree of irregularity in the fluctuations of a time series by calculating the likelihood that a vector of $m$ data points matches a template (of equal length) within a tolerance range of $r$ normalized to the SD of the signal [37] (Eq 2),

$$CMSE = \frac{1}{\tau}\sum\nolimits_{k=1}^{\tau} SampEn(\gamma_k^{\tau}, m, r) \qquad (2)$$

where $k$ is the number of coarse-grained time-series for a scale factor of $\tau$. We set $r$ to 0.2 and $m$ to 2. The series corresponding to the different time-scales were constructed by coarse graining the original time series using a moving average with non-overlapping windows of the size (i.e., number of points) of the scale factor, with scale 1 being the original time series. To obtain smoother curves, we applied the procedure introduced by Wu et al. (2013) consisting for the $n^{th}$ scale of averaging $n$ coarse grained signals obtained by shifting by one point $n$ times the moving average window (for an illustration see [33]). Classical MSE was also computed, but not reported, to assure that the obtained curves with the modified coarse-graining were faithful to the original method.

SampEn was calculated for the original and the coarse-grained time series for 25 scales ensuring thereby reliable entropy estimations. The MSE curves were obtained by plotting SampEn values as a function of the scale factor. To capture their general characteristics, and therefore the complexity of the studied signals, we analyzed the mean SampEn of each curve. To be noted that our low-pass filtering at 30 Hz would only interfere with the entropy values of the first four scales. Nevertheless, they were not excluded because they had no impact on the observed results.

**Dynamics characterization.** The dynamics of force production can be decomposed into a deterministic and a stochastic component. The deterministic and stochastic components of

the force production dynamics were quantified by applying the Kramers-Moyal expansion to conditional probability matrices (cf., [28,38,39]). The conditional probability matrix describes the probability to find the system in a given state at a time $t + \Delta t$ given its earlier state at time $t$.

As in Vieluf et al., (2017), the decomposition of applied force into deterministic and stochastic components was done on the Hilbert phase transformed data for the sinusoidal force modulation data (i.e., the phase angle). For each trial, for both tasks separately, the two-dimensional conditional probability matrix P(AF', $t + \Delta t$|AF, $t$), which denotes the probability to find the system at state AF' at time $t + \Delta t$ given its applied force state AF at an earlier time step $t$, was computed using a bin size of (5 x SD(AF))/n with n = 7. The range of the AF space sampled was from -2.5 x SD to 2.5 x SD for the force maintenance task, and the range of the AF space sampled was from -pi to pi sinusoidal force modulation task. $\Delta t$ was set to one sample, i.e., 4.17 ms. Then, for each condition and participant the average conditional probability matrix across all trials was computed. The deterministic and stochastic dynamics (also referred to as drift and diffusion coefficients) were next calculated based on P(AF',$t + \Delta t$ | AF,$t$) as:

$$D^n = \lim_{\Delta t \to 0} \frac{1}{\Delta t} \int \frac{(AF' - AF)^n}{n!} P(AF', t + \Delta t|AF, t)dx' \qquad (3)$$

The deterministic (drift) and stochastic (diffusion) dynamics are obtained for $n$ = 1 and 2, respectively. To evaluate the fixed point's stability in the force maintenance task, the slope of the deterministic component (the drift coefficient) across the three middle bins (i.e., where the coefficient changed sign; Fig 4) was determined by a linear regression.

### Statistical analyses

The force-frequency interactions were analyzed using repeated-measures ANOVA on the performance variables (mean and SD of force, relative phase, relative phase variability, and mean SampEn) with as between-subject factor age group [young, older adults], and force level and frequency as within-subjects factors. The same analysis was conducted for the drift and diffusion coefficients, but with bins as an additional factor. A one-way ANOVA tested whether the slopes of the drift coefficients for the force maintenance task differed between the age groups. Effect sizes are reported as the partial eta squared ($\eta_p^2$). Whenever sphericity was violated, the Greenhouse-Geisser correction was applied. To decompose significant effects post hoc, the Bonferroni test was *used. The level of significance* was set to $\alpha$ = 0.05. Statistical analyses were conducted using the SPSS 24.0 statistical package (IBM). For sake of brevity, only significant effects are detailed in the results section.

## Results

For an overview, a general summary of the significant main effects and interactions revealed by the analyses of variance can be found in Table 1.

### MVC

Young participants' MVC increased ($t$(53) = -2.45, $p$ = .018) by a small amount (7%) in the post-test (66.72±16.59 N) relative to the pre-test (61.82±16.68 N). For the elderly, however, the MVC in the pre-test (54.19±12.30 N) and post-test (51.14±17.04 N) were not significantly different ($t$(35) = 1.38, $p$ = .18). Overall, the MVC of the young participants was significantly higher than that of the elderly ($p < .001$). The absence of a decrease in MVC following the experimental conditions allowed us to verify that our procedure did not induce fatigue in any of the two groups.

**Table 1. Repeated measures ANOVA table of the significant results in the constant and sinusoidal force tracking tasks.** The variables that were analyzed were the mean applied force, standard deviation (SD), coefficient of variation (CV), Root-Mean-Square-Error (RMSE), multi-scale entropy (MSE), drift and diffusion coefficients.

| Variable | | Constant force | | | |
|---|---|---|---|---|---|
| | | **F** | **df** | **p** | **$\eta_\pi^2$** |
| **Mean force** | Force | 227308 | 1,28 | <.001 | 1 |
| | Force×Age | 8.85 | 1,28 | <.01 | 0.24 |
| **SD** | Force | 156.98 | 1,28 | <.001 | 0.85 |
| | Age | 6.26 | 1,28 | .018 | 0.18 |
| **CV** | Force | 11.76 | 1,28 | .002 | 0.3 |
| | Age | 5.67 | 1,28 | .02 | 0.17 |
| **RMSE** | Force | 72.79 | 1,28 | <.001 | 0.72 |
| | Age | 9.99 | 1,28 | .004 | 0.26 |
| **MSE** | Age | 10.38 | 1,28 | .003 | 0.27 |
| **Drift coefficient** | Bins | 246.94 | 1,30 | <.001 | 0.9 |
| | Force×Bins | 89.6 | 1,39 | <.001 | 0.76 |
| **Diffusion coefficient** | Force | 59 | 1,28 | <.001 | 0.68 |
| | Bins | 40.64 | 1,34 | <.001 | 0.59 |
| | Force×Bins | 27.67 | 1,40 | <.001 | 0.5 |
| | | Sinusoidal force | | | |
| | | F | df | p | $\eta_\pi^2$ |
| **Mean force** | Force | 35544 | 1,28 | <.001 | 0.99 |
| | Frequency | 6.26 | 1,28 | <.001 | 0.41 |
| | Force×Age | 27.46 | 1,28 | <.001 | 0.5 |
| | Age×Frequency | 15.81 | 1,28 | <.001 | 0.36 |
| | Frequency×Force | 7.47 | 1,28 | .01 | 0.21 |
| **Phase locking** | Age | 4.54 | 1,28 | .04 | 0.14 |
| **Circular variance** | Age | 71.8 | 1,28 | <.001 | 0.72 |
| | Force | 24.61 | 1,28 | <.001 | 0.47 |
| | Frequency | 47.72 | 1,28 | <.001 | 0.63 |
| | Age×Force | 11.91 | 1,28 | .002 | 0.3 |
| | Force×Frequency | 37.7 | 1,28 | <.001 | 0.57 |
| | Age×Force×Frequency | 1.84 | 1,28 | <.001 | 0.38 |
| **RMSE** | Age | 118.67 | 1,28 | <.001 | 0.81 |
| | Force | 5.87 | 1,28 | .02 | 0.17 |
| | Frequency | 21.21 | 1,28 | <.001 | 0.43 |
| | Force×Frequency | 10.96 | 1,28 | .003 | 0.28 |
| | Age×Force×Frequency | 8.11 | 1,28 | .008 | 0.23 |
| **MSE** | Age | 11.66 | 1,28 | .002 | 0.29 |
| | Force | 45.53 | 1,28 | <.001 | 0.61 |
| | Frequency | 230.31 | 1,28 | <.001 | 0.89 |
| | Age×Force | 8.72 | 1,28 | .006 | 0.24 |
| | Age×Frequency | 5.96 | 1,28 | .02 | 0.18 |
| **Drift coefficient** | Force | 6.57 | 1,28 | .02 | 0.19 |
| | Frequency | 711.82 | 1,28 | <.001 | 0.96 |
| | Bins | 215.9 | 4,101 | <.001 | 0.89 |
| | Age×Frequency | 7.45 | 1,28 | .01 | 0.21 |
| | Age×Bins | 9.8 | 3,101 | <.001 | 0.26 |
| | Force×Bins | 5.19 | 5, 128 | <.001 | 0.16 |
| | Age×Force×Bins | 4.74 | 5,128 | .001 | 0.13 |
| | Frequency×Bins | 24.38 | 4,121 | <.001 | 0.47 |
| | Age×Frequency×Bins | 4.08 | 4,121 | .003 | 0.13 |

*(Continued)*

**Table 1.** (Continued)

| Variable | | Constant force | | | |
|---|---|---|---|---|---|
| | | *F* | *df* | *p* | $\eta_p^2$ |
| **Diffusion coefficient** | *Age* | 18.68 | 1,28 | <.001 | 0.4 |
| | *Force* | 15.32 | 1,28 | .001 | 0.35 |
| | *Frequency* | 950.89 | 1,28 | <.001 | 0.97 |
| | *Bins* | 232.63 | 3,103 | <.001 | 0.89 |
| | *Age×Bins* | 17.07 | 3,103 | <.001 | 0.38 |
| | *Force×Frequency* | 4.29 | 1,28 | .048 | 0.13 |
| | *Force×Bins* | 8.53 | 5,143 | <.001 | 0.23 |
| | *Age×Force×Bins* | 2.89 | 5,143 | .02 | 0.094 |
| | *Frequency×Bins* | 26.33 | 4,115 | <.001 | 0.49 |
| | *Age×Frequeny×Bins* | 4.13 | 4,115 | .003 | 0.13 |
| | *Force×Frequency×Bins* | 2.87 | 4,111 | .03 | 0.09 |

## General characteristics of force performance

**Constant force.** As required by the task, mean applied force in % MVC differed significantly between the force levels ($F(1,28) = 227308$, $p < .001$, $\eta_p^2 = 1.00$; Fig 2A), but not between age groups. Force level interacted with age group ($F(1,28) = 8.85$, $p < .01$, $\eta_p^2 = .24$; Fig 2A). The decomposition of the interaction showed however that mean force increased from the 10% to the 30% MVC for both young and elderly (both young and elderly: $p < .001$), with no statistically significant differences between groups.

The within participants mean standard deviation differed significantly between force levels ($F(1,28) = 156.98$, $p < .001$, $\eta_p^2 = .85$; Fig 2C), and between age groups ($F(1,28) = 6.26$, $p = .018$, $\eta_p^2 = .18$), with no statistically significant interaction between the two factors. The decomposition of the main effects revealed that the SD was larger in elderly than in young adults (10% MVC: $p = .03$, 30% MVC: $p = .02$, grouped: $t(58) = -2.6$, $p = .01$), and in both groups for the 30% than for the 10% MVC condition (YN: $p < .001$, EL: $p < .001$; grouped: $t(58) = -5.7$, $p < .001$).

The coefficient of variation (CV; Fig 2E) was significantly affected by force level ($F(1,28) = 11.76$, $p = .002$, $\eta_p^2 = .30$) and age group ($F(1,28) = 5.67$, $p = .02$, $\eta_p^2 = .17$), but no interaction effect was found. The decomposition of the main effects revealed that the CV significantly decreased with increasing force for young adults ($p = .001$), but not for elderly. The CV was significantly higher for older adults than for young adults at both force levels (10% MVC, $p = .04$; 30% MVC, $p = .02$).

The root mean squared error (RMSE) differed between young and elderly participants ($F(1,28) = 9.99$, $p = .004$, $\eta_p^2 = .26$; Fig 3A), and showed an effect of force level ($F(1,28) = 72.79$, $p < .001$, $\eta_p^2 = .72$; Fig 3A). It was larger for elderly than for young participants ($t(58) = -3.3$, $p = .002$), and for the high force level than for the low force level ($t(58) = -4.93$, $p < .001$).

**Sinusoidal force.** The mean applied force differed between force levels ($F(1,28) = 35544$, $p < .001$, $\eta_p^2 = .99$, Fig 2B) and between frequencies ($F(1,28) = 6.26$, $p < .001$, $\eta_p^2 = .41$), but there was no main effect of age groups. The interactions between age group and force level ($F(1,28) = 27.46$, $p < .001$, $\eta_p^2 = .50$), age group and frequency ($F(1,28) = 15.81$, $p < .001$, $\eta_p^2 = .36$), and frequency and force ($F(1,28) = 7.47$, $p = .01$, $\eta_p^2 = .21$) reached significance. Post-hoc tests revealed that the mean applied force increased with force level (all $p < .001$, grouped: $t(118) = -106.7$, $p < .001$), but the frequency effect did not reach significance. Only the 30% MVC at 0.8Hz set apart the young from the older adults ($t(28) = 3.4$, $p = .002$), where the

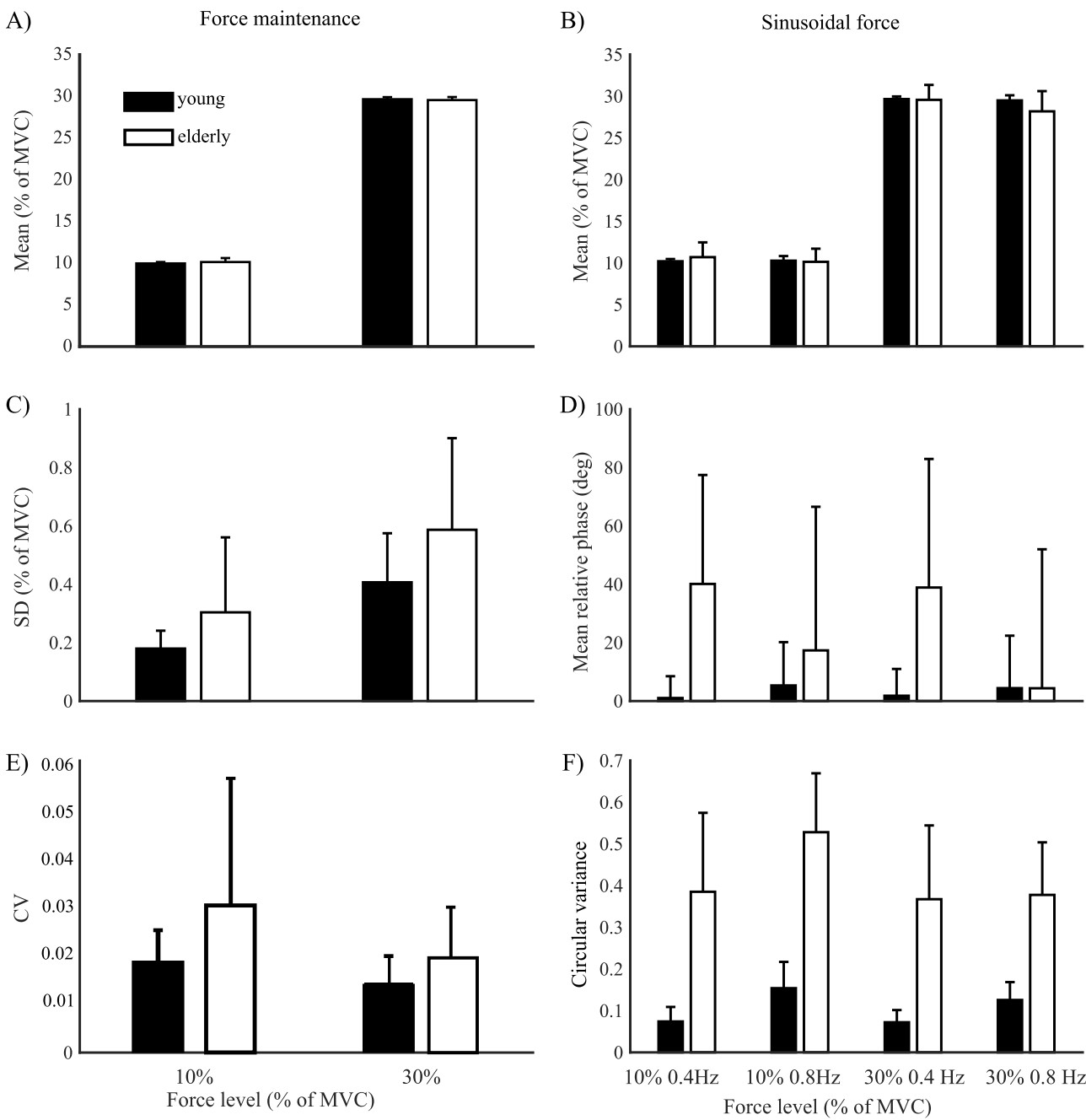

**Fig 2. General characteristics of force production.** (A) Mean force in the constant task and (B) the modulation task. (C) Standard deviation for the constant task. (D) Mean relative phase for the modulation task. (E) Coefficient of variation for the constant task. (F) Uniformity for the modulation task. Black bars represent the results of the young group, and white bars those of the elderly. Error bars represent the standard deviation.

elderly showed a significantly lower mean applied force (young adults: 29.49±0.62%MVC, older adults: 28.18±2.42%MVC).

Elderly showed statistically significant difference from young participants in synchronizing their movements with the target force ($F(1,28) = 4.54$, $p = .04$, $\eta_p^2 = .14$, Fig 2D). Neither force level nor frequency effects on the phase locking reached statistical significance. Yet, from Fig 2D it can be seen that the mean difference between the target force and the produced force was

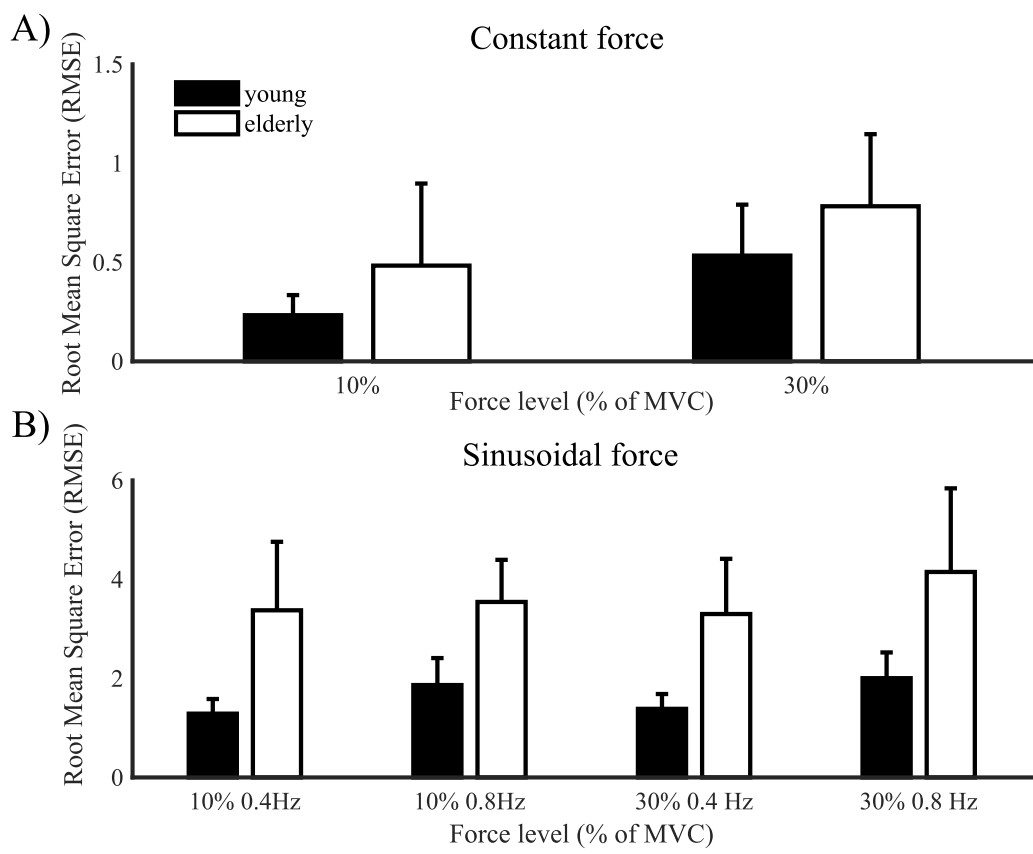

**Fig 3. Root mean square error (RMSE) of produced force.** RMSE at 10% and 30% maximum voluntary contraction for young (black) and elderly (white) participants in the (A) constant force conditions and (B) the modulation force conditions. Error bars represent the standard deviation.

significantly higher for elderly than for young participants ($t(118)$ = -3.33, $p$ = .001), indicating that elderly had difficulties following the sinusoidal target.

The circular variance was significantly different for elderly as compared to young participants ($F(1,28)$ = 71.80, $p <$ .001, $\eta_p^2$ = .72, Fig 2F), and revealed further effects of force level ($F(1,28)$ = 24.61, $p <$ .001, $\eta_p^2$ = .47) and frequency ($F(1,28)$ = 47.72, $p <$ .001, $\eta_p^2$ = .63). Interactions were found between the age group and force level ($F(1,28)$ = 11.91, $p$ = .002, $\eta_p^2$ = .30), force level and frequency ($F(1,28)$ = 37.70, $p <$ .001, $\eta_p^2$ = .57), and between all three factors ($F(1,28)$ = 1.84, $p <$ .001, $\eta_p^2$ = .38). Post-hoc tests showed that the circular variance increased with age (all comparisons $p <$ .001), but grouped frequency and force level did not reach statistical significance. The variance increased for the young participants at high frequencies for 10% and 30% MVC ($p <$ .001) compared to the low frequencies, whereas for the more variable elderly these effects were not significant.

The RMSE was influenced by age group ($F(1,28)$ = 118.67, $p <$ .001, $\eta_p^2$ = .81; Fig 3B), with post-hoc decomposition showing higher values for elderly than for young participants ($t(116)$ = -14.9, $p <$ .001, Fig 3B). A statistically significant effect was also found for both the force level and the frequency factors (force: $F(1,28)$ = 5.87, $p$ = .02, $\eta_p^2$ = .17, frequency: $F(1,28)$ = 21.21, $p <$ .001, $\eta_p^2$ = .43) that interacted with each other ($F(1,28)$ = 10.96, $p$ = .003, $\eta_p^2$ = .28). Furthermore, frequency, force and age group interacted ($F(1,28)$ = 8.11, $p$ = .008, $\eta_p^2$ = .23). Effects' decomposition showed that the RMSE increased at the high frequency in

young participants at both the 10% ($t(34)$ = -5.65, $p$ < .001) and 30% MVC ($t(34)$ = -5.84, $p$ < .001), with no statistically significant differences revealed in elderly.

## Multi scale entropy

**Constant force.** In the constant force condition, the statistical analysis showed that the mean SampEn was affected by age group ($F(1,28)$ = 10.38, $p$ = .003, $\eta_p^2$ = .27), but not by force level. It was higher for young as compared to elderly participants at both force levels ($t(58)$ = 4.43, $p$ < .001; Fig 4A). This also became apparent in MSE curves (Fig 4C), where an increase of SampEn was structurally visible for young as compared to older adults, especially so at longer (coarser) scales.

**Sinusoidal force.** In the sinusoidal force condition there was also a statistically significant effect of age group ($F(1,28)$ = 11.66, $p$ = .002, $\eta_p^2$ = .29), but here the SampEn was higher for the elderly as compared to the young adults ($t(118)$ = -3.77, $p$ < .001, Fig 4B and 4D). Statistically significant effects were as well found for the factors force level ($F(1,28)$ = 43.53, $p$ < .001, $\eta_p^2$ = .61), and frequency ($F(1,28)$ = 230.31, $p$ < .001, $\eta_p^2$ = .89). Age group interacted with force level ($F(1,28)$ = 8.72, $p$ = .006, $\eta_p^2$ = .24), and with frequency ($F(1,28)$ = 5.96, $p$ = .02, $\eta_p^2$ = .18). Post-hoc tests revealed a tendency for a three-way interaction ($p$ = .06), showing that for older adults, mean SampEn was not affected by force level when the frequency

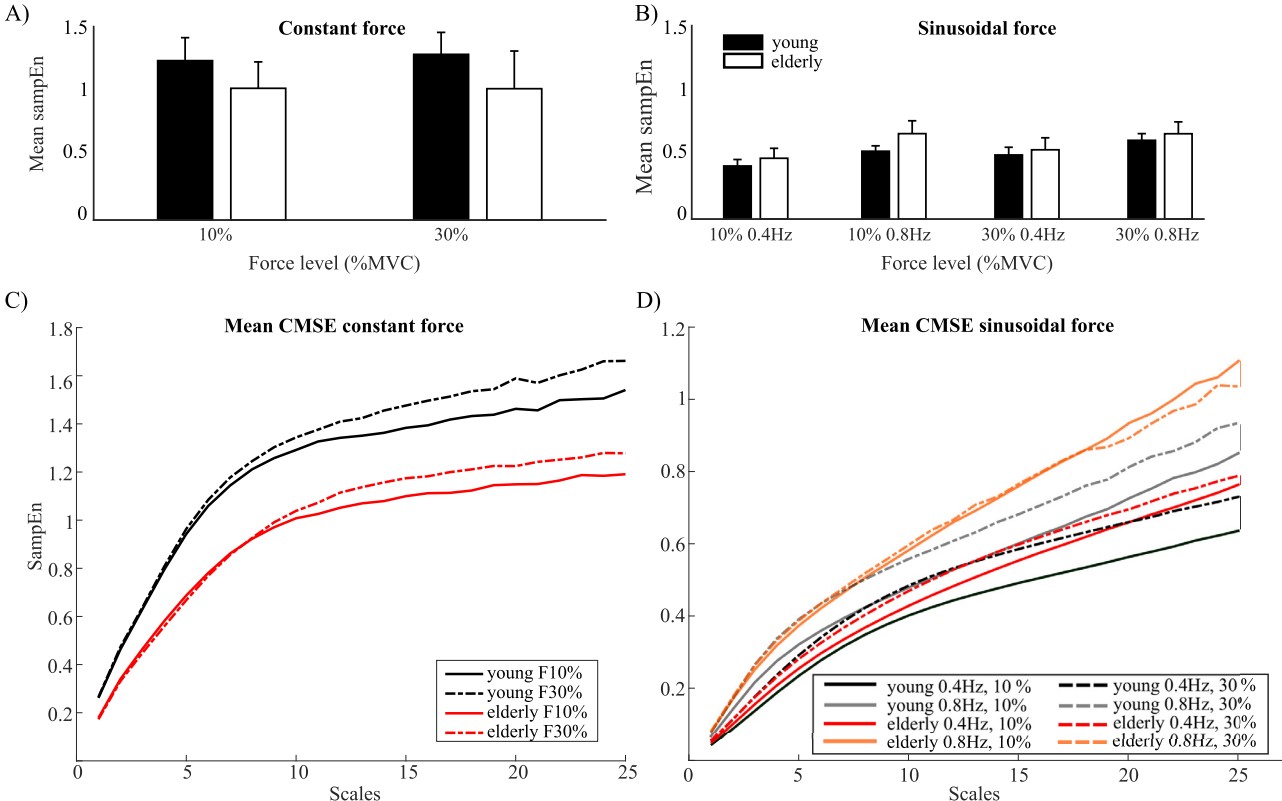

**Fig 4. Multiscale entropy of produced force.** Mean sample entropy for (A) constant and (B) modulation tasks, for young (black) and elderly participants (white) for 10% and 30% maximum voluntary contraction (MVC) and at 0.4 and 0.8 Hz. Error bars represent the standard deviation. The mean composite multiscale entropy curves for young (black) and elderly (red) at (C) constant forces of 10% MVC (filled line) and 30% MVC (dotted line). The mean composite multiscale entropy curves for young (black and grey) and elderly (red and orange) at (D) sinusoidal forces at 10% MVC (filled line) at 0.4 and 0.8 Hz and at 30% MVC (dotted lines).

remained the same. All other comparisons between age group, force, and frequency reached statistical significance (all $p < .001$, except for the frequency comparison at 30% MVC for elderly: $p < .005$). Thus, the mean SampEn was higher for older adults than for young adults, but the impact of force and frequency levels on the entropy changed with age, with task-constraints modulating mainly young participants' behavior.

### Drift and diffusion coefficients

**Constant force.** The drift coefficients were found to be statistically different across the bins ($F(1.11, 30.55) = 246.94$, $p < .001$, $\eta_p^2 = .90$), with a significant interaction of force and bin ($F(1.40, 39.10) = 89.60$, $p < .001$, $\eta_p^2 = .76$). All bins differed from one another ($p < .001$), except for the comparisons with the bin in the center (i.e., bin 3). For the drift (i.e., deterministic) component, the curve across bins shows a nearly straight line that crosses the horizontal axis at 0, indicating the presence of a fixed point attractor at that force level (Fig 5). A linear regression on the drift coefficients for 10% and 30% MVC showed a steeper negative slope at 30% MVC ($F(1,28) = 251.06$, $p < .001$, $\eta_p^2 = .79$; young: $b = -3.53$, elderly: $b = -3.92$, Fig 5) than at 10% MVC (young: $b = -1.51$, elderly: $b = -1.75$, Fig 5). The slopes were not however statistically different between the young and elderly. Thus, in sum, force maintenance was more stable in the 30% MVC condition than in the 10% MVC condition, but the stability did not differ significantly between both age groups.

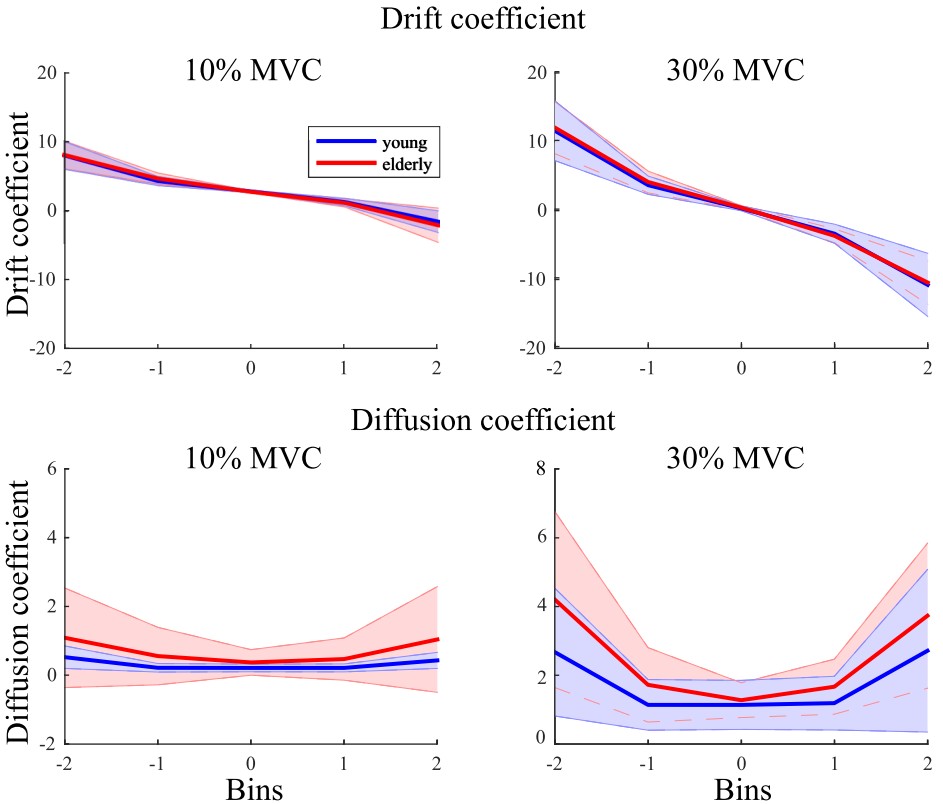

**Fig 5. Drift and diffusion coefficients for a constant force task at 10% and 30% maximum voluntary contraction for young (blue) and elderly (red).** The shaded area represents the standard deviation.

The diffusion coefficients were significantly influenced by force level ($F(1,28) = 59.00$, $p < .001$, $\eta_p^2 = .68$), but they were not significantly influenced by age group. The diffusion coefficient changed over bins ($F(1.23, 34.53) = 40.64$, $p < .001$, $\eta_p^2 = .59$). There was a statistically significant interaction between force level and bins ($F(1.43, 40.05) = 27.67$, $p < .001$, $\eta_p^2 = .50$). Post-hoc analysis revealed that the diffusion coefficients for all matching bins were significantly lower for 10% compared to 30% MVC ($p < .05$). In other words, the degree of stochasticity was lower in the 10% MVC condition than in the 30% MVC condition, but was statistically similar for both age groups.

**Sinusoidal force.** The drift coefficients plotted as a function of bins based on the Hilbert phase revealed a bimodal structure with no zero crossing of the horizontal axis (Fig 6A). The force level ($F(1,28) = 6.57$, $p = .02$, $\eta_p^2 = .19$) and frequency ($F(1,28) = 711.82$, $p < .001$, $\eta_p^2 = .96$) had a significant influence on the drift coefficients. Drift coefficients changed over bins ($F(3.6,101.0) = 215.90$, $p < .001$, $\eta_p^2 = .89$). No main effect of age group was found on the drift coefficients. Interaction effects were found between frequency and age group ($F(1,28) = 7.45$, $p = .01$, $\eta_p^2 = .21$), bins and age group ($F(3.6,101.0) = 9.80$, $p < .001$, $\eta_p^2 = .26$), force level and bins ($F(4.6, 127.7) = 5.19$, $p < .001$, $\eta_p^2 = .16$), force, bins, and age group ($F(4.6, 127.7) = 4.74$, $p = .001$, $\eta_p^2 = .13$), frequency and bins ($F(4.3, 121.3) = 24.38$, $p < .001$, $\eta_p^2 = .47$), and frequency, bins and age groups ($F(4.3, 121.3) = 4.08$, $p = .003$, $\eta_p^2 = .13$). The drift coefficients for the low frequency were significantly lower than for the high frequency conditions (all bins $p < .001$), irrespective of force level. This (trivial) effect simply reflects that the low frequency condition was performed slower than the high frequency condition. The bins 1, 2, 3, 6 and 7 did not differ between 10% and 30% MVC when the frequencies were grouped. Older adults showed a statistically significant difference from young adults for bins 3 and 8 at 10% MVC at 0.4 Hz, bins 3 and 6 at 30% MVC at 0.4 Hz, bin 6 at 10% at 0.8 Hz, and bin 1 at 30% MVC at 0.8 Hz ($p < .05$). These effects are most likely due to the age group difference in the (mean) relative phase between the stimulus and produced force; after all, a higher drift coefficient at a given bin reflects the faster rate of phase change at the corresponding phase. Most importantly, the observed profiles unambiguously ruled out a fixed point dynamics and are suggestive of a limit cycle dynamics, and that this observed dynamics was independent of age.

For the diffusion coefficients (Fig 6B), all the four factors, that is, age group ($F(1,28) = 18.68$, $p < .001$, $\eta_p^2 = .40$), force level ($F(1,28) = 15.32$, $p = .001$, $\eta_p^2 = .35$), frequency ($F(1,28) = 950.89$, $p < .001$, $\eta_p^2 = .97$), and bins ($F(3.7,103.2) = 232.63$, $p < .001$, $\eta_p^2 = .89$) had a statistically significant influence. Interaction effects were found between the bins and age group ($F(3.7,103.3) = 17.07$, $p < .001$, $\eta_p^2 = .38$), force and frequency ($F(1,28) = 4.29$, $p = .048$, $\eta_p^2 = .13$), force and bins ($F(5.1,143.3) = 8.53$, $p < .001$, $\eta_p^2 = .23$), force, bins and age group ($F(5.1,143.3) = 2.89$, $p = .02$, $\eta_p^2 = .094$), frequency and bins ($F(4.1,115.0) = 26.33$, $p < .001$, $\eta_p^2 = .49$), frequency, bins and age groups ($F(4.1,115.0) = 4.13$, $p = .003$, $\eta_p^2 = .13$), and force, frequency and bins ($F(3.97,111.0) = 2.87$, $p = .03$, $\eta_p^2 = .09$). For the 0.8 Hz conditions, the diffusion coefficients for bins 3, 4 and 8 were higher for older adults than for young adults ($p < .05$). For the 0.4 Hz conditions, a similar effect was found for bins 2, 3, 4, 6 and 7 ($p < .05$) at both 10% and 30% MVC, and additionally for bin 8 at 10% MVC ($p < .001$). The diffusion coefficients for the low frequency were significantly lower than for the high frequency conditions (all bins $p < .001$), regardless of force level. The bins 2, 3, 6 and 7 did not significantly differ between 10% and 30% MVC when the frequencies were grouped. The results for the diffusion coefficients resemble the results for the drift coefficients with the notable exception that, in this case, the age group effect was significant: the elderly performance was noisier than that of the young participants.

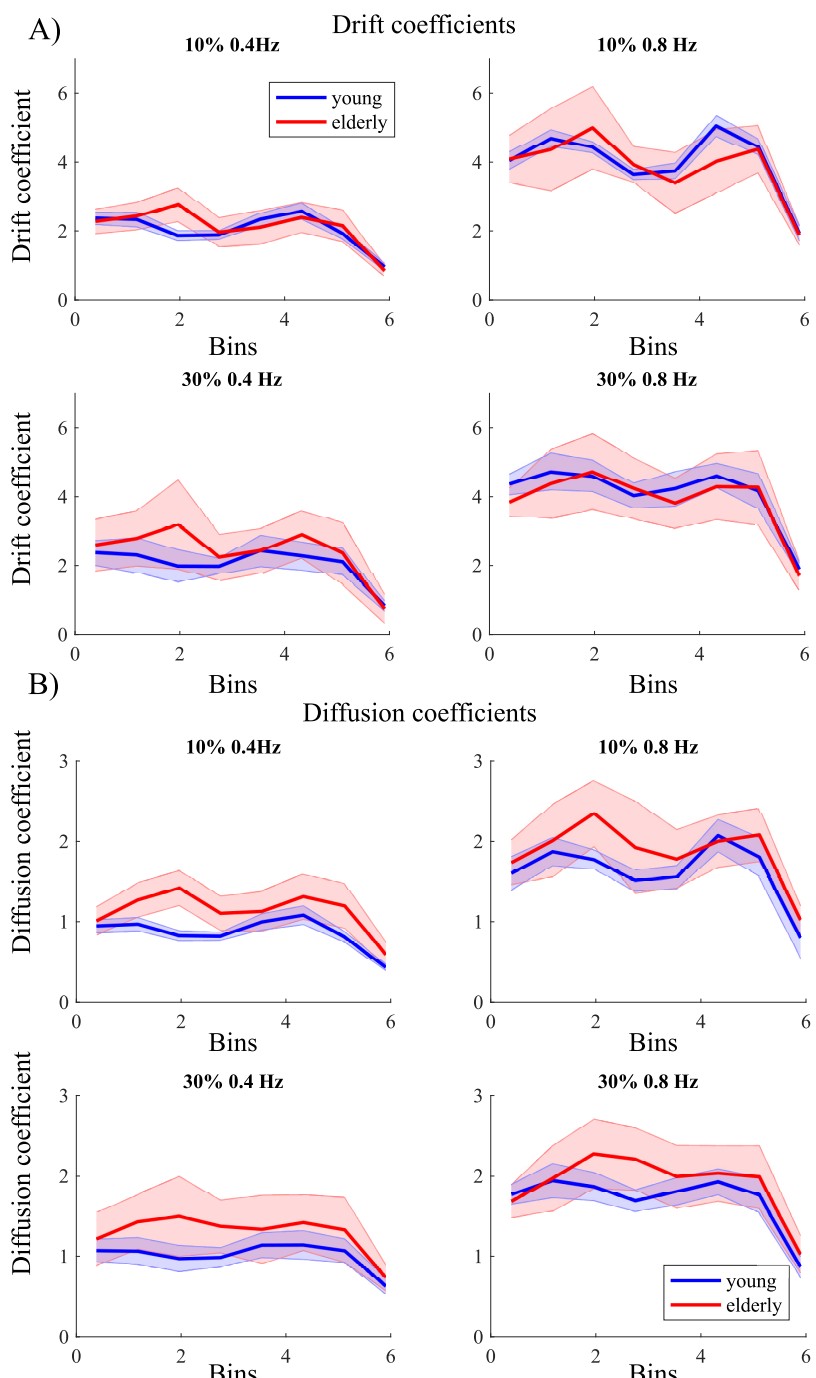

**Fig 6. Drift and diffusion coefficients for the modulation task.** (A) Drift and (B) diffusion coefficients for 10% and 30% maximum voluntary contraction for a frequency 0.4Hz and 0.8Hz for young (blue) and elderly (red) over bins. The shaded area represents the standard deviation.

## Discussion

Our study investigated the effects of aging and task constraints on force control in a visuomotor task that involved on-screen tracking of either a constant or a sinusoidal force target, at different force levels and force modulation frequency. This chosen task involves multiple control

processes (e.g., force production, visuomotor integration, and attention) that are implicated in daily living activities, and presents the advantage of being easy to implement and undergo for both research and clinical purposes. To offer a thorough characterization of the produced sensorimotor behavior in young and elderly participants, we combined measures of performance, complexity, and dynamics. In the following sections, we discuss our main findings and the contribution of the different metrics to the understanding of how the aging system adapts to changes in task constraints.

### Force accuracy and variability

**Force tracking tasks are more demanding for elderly than for young adults.** Although the force modulation task evoked an increase in force variability in both groups, older adults struggled more than young adults in tracking the cyclic target. Specifically, they presented a phase lag, deviated more from the target force (an increase of 121.9% RMSE relative to young participants), and were more variable than young adults (an increase of 289.3% of circular variance relative to young participants). The elderly were also more variable than the young adults in the constant task, although to a much lesser extent compared to the modulation task (+60.8% of the RMSE, +56.1% of the CV, and +54.6% of the SD as compared to young adults). These results are consistent with those reported in previous studies, showing that age effects are more pronounced in the arguably more demanding force modulation tasks than in force maintenance tasks [3,40,41]. Visual processing demands on the neuromuscular system might be dependent on the nature of task, with a higher load incurred by a varying force target. This assumption could explain the increased difficulties encountered by the elderly in the modulation task, knowing that visuomotor information processing is compromised with aging [42–44]. However, this cannot be tested through our design and a strict comparison between tasks is limited due to their different nature and the different measures accounting for the produced behavior in each.

**The elderly comply with relative target force levels.** Older adults have been reported to overshoot target forces [3,45] or grip forces [46], possibly as means of compensating for the loss of tactile sensitivity [47,48]. In our experiment where we set relative (%MVC) target force levels, instead of absolute levels as in the aforementioned studies, we did not observe any overshooting behavior. Adjusting the target force to individual force capacities might have made the task more accessible to the elderly. This result can be implemented in gerontechnologies to compensate some of age-related deficits and limit their functional consequences.

**Poorer force tracking in elderly independently of task conditions.** Previous studies hinted at a possible weaker coupling between the applied force and the target force with age [3,25]. Our older adults group showed clearly a larger and more variable phase lag between the target force and the applied force than young adults. The pervasive larger phase lags in elderly could be indicative of perceptual and perceptuomotor deficits, whereas the greater variability could be a sign of a weaker coupling with the task/environment, rendering the behavior less efficient and less stable [49–53]. Indeed, young adults were more precise and consistent, and were only destabilized when force modulation frequency increased. Although we cannot unambiguously determine the origin of the reduced accuracy in elderly, it seems most likely that the coupling between the applied force (action) and the target force (perception) weakens with aging. The prevailing view linking age-related motor control deficits to increased white noise in the perceptual-motor system ([54], i.e., the neural noise hypothesis [55]) is however challenged by the investigation of the dynamics and the time-scale dependent structure of the force signal, which offers a different perspective putting forward the interplay between internal

and external constraints [56–58]. For that reason, in the following, we examine the processes contributing to the decreased ability of the elderly to adapt to the constraints of the task.

## Complexity of force fluctuations

The two used task modalities impose different constraints on the sensorimotor system. As Vaillancourt and Newell (2002) conceptualized it, the constant task requires the stabilization around a target line with the sensorimotor system being more or less free to express its intrinsic fluctuations, whereas the modulation task is dominated by the production of a perfectly regular pattern, restricting thereby furthermore behavioral fluctuations to a dominant time-scale [24].

The results of the MSE analysis showed that the generally poorer performance in elderly was associated with a distinctive pattern of changes in force fluctuations. In line with the predictions of Vaillancourt and Newell (2002), the older adults' force fluctuations were less complex (more predictable with lower entropy across most of the time-scales) than those of the young adults in the constant force production task, and the reverse in the sinusoidal task (more complex). These results extend those of Vaillancourt and Newell (2003) who used a mono-scale entropy measure (approximate entropy) and could thus formally only infer changes in behavioral irregularity rather than complexity. The obtained MSE curves also show the locus of the age group differences at coarser scales (> scale factor 10), reflecting mostly low frequencies related to sensorimotor processing in force control (0-12Hz [59]), for further development in force control). Vaillancourt, Sosnoff and Newell (2004) argued that age-related changes in complexity are dependent on task dynamics. In line with the aging induced loss of complexity and dedifferentiation discussion developed in Sleimen-Malkoun et al. (2014), the observed task-dependent effect of aging on behavioral complexity can be interpreted as a reduced dynamical range of the neurobehavioral repertoire. This means that the aged neuro-musculo-skeletal system has a poorer intrinsic connectivity within and between the functional domains, and is furthermore less proficient in assembling sensorimotor synergies meeting task constraints in challenging contexts. Said differently, young adults express a complex behavior that reflects the richness of their underlying degrees of freedom (dof) when the task allows it, but can successfully compress their dof to become more regular when the task requires it.

Indeed, all conditions taken together, one can see that, when going from constant to sinusoidal force production, younger participants reduced on average their mean MSE by 61.7%, while the elderly did so by only 47.5%. Interestingly, a similar account can be made when considering task constraints in terms of force level and frequency. Although aging × force level interaction did not reach significance in the constant task, aging interacted with force level and frequency, with also a tendency to a three-way interaction in the sinusoidal (considered the more challenging) task. In this latter context, young adults' behavioral complexity scaled-up when task constraints increased (higher force, higher frequency). It could be suggestive of a less efficient sensorimotor control echoing the increase in behavioral variability and decrease in accuracy. In older adults, no significant effect of force level was found, suggesting that their complexity was comparable at both levels. This said, elderly are often reported to have more trouble controlling lower forces [14–17]. Conversely, increasing the target frequency altered significantly the time structure of force control, pushing it further away from the regularity imposed by the task, more so for the lower (10%) than the higher (30%) force level.

It is noteworthy that the classical distributional measures of performance, which presented a very important inter-individual variability, failed to convey this distinction between task conditions, which hints at frequency as a candidate control parameter of the dynamics. This issue will be developed in the next section, wherein we explore the signature of the underlying

stochastic and deterministic components, as well as attempt to link the dynamics to the above-discussed performance and complexity changes under task-related and aging-related constraints.

## Dynamical signatures of force maintenance and force modulation

Force control has been shown to result from the conjunction of deterministic and stochastic dynamical components, which show different signatures depending on the nature of the task (sinusoidal force modulation or constant force maintenance) with a reduced effect of expertise and age in late middle-aged [3]. What remains unanswered is whether the task-dependent changes in dynamics would be invariant under different force levels and different force modulation frequencies, and whether aging would have a more pronounced effect in late adulthood.

For the force maintenance task, we found a clear fixed-point dynamic regardless the applied force level. The steepness of the curve around the fixed point increased with force level, indicating that the large force deviations relative to the required force are counteracted with more vigor than for the lower force level. However, this finding contradicts the results reported in Frank et al. (2006), who showed a decrease of the slope of the deterministic component with force level up to 40% (i.e., for force level 10%, 20% and 40%, but not for 60% and 70% MVC). This deviation may well be due to a difference in normalization method: for each participant, Frank et al. normalized the diffusion coefficient by dividing it by the average diffusion coefficient across all force levels. We refrained from doing so in order to be able to examine corresponding force fluctuations in terms of the actual forces produced. Regardless, as expected, the stochastic fluctuations were the lowest around the fixed point [3,28], and they increased with force level.

For the cyclic force modulation task, the underlying dynamics was previously identified as of oscillatory nature, most likely stemming from a limit-cycle generating mechanism [3]. Two local maxima and minima were identified that were associated with the fast ascending and descending phases of force production around the force maxima and minima, respectively. The question came up whether this oscillatory behavior might change to a bistable fixed point dynamics at lower frequencies, analogous to the bifurcation seen in trajectory production tasks (cf., [60,61]). Here, in accordance with Vieluf et al. (2017), no zero-crossing was observed in the force modulation task. Furthermore, the peaks associated with the fast descending and ascending phase were more pronounced at the higher frequency. Thus, at the lower frequency the force produced evolved more harmonically and, consequently, increased and decreased more smoothly than at the higher frequency. In other words, contrary to our expectation, the dynamics at the higher frequency was more nonlinear, and therefore appeared closer to a bifurcation than that at the lower frequency. In that regard, our expectation was based on multiple observations that increasing movement frequency in (spatially unconstrained) trajectory production tasks may evoke a bifurcation from a regime involving one (or more) stable fixed point(s) to a limit cycle (i.e., oscillatory) regime (cf., [60,61]). Our present results suggest that, if existent, a transition between regimes in force production evolves in the opposite direction as a function of movement frequency. Having said this, the magnitude of the drift coefficients increased with frequency, which (also) suggests that at lower frequencies eventually the zero-line will be touched, and thus fixed points will exist. If, however, across bins the drift coefficients 'flatten out' (become a straight line) as they approach zero, all coefficients (i.e., at all bins) would become zero simultaneously. Under this scenario, task performance would break down, and it can therefore be expected that before that point a distinct dynamical organization would be assembled ensuring continued task performance.

Overall, we have no proof that our experimental manipulations brought the force control system close to a bifurcation, but solely an indication that the dynamics became more nonlinear with increasing movement frequency in the modulation task. The bifurcation parameter has yet to be identified for the force control system. It seems though that the force modulation frequency could play this role rather than the produced force level. This issue is of particular interest to follow up as our results suggest that the neuromuscular system might operate differently when controlling force in the absence of movement (as in our study) than when producing a movement trajectory [60,61]. This need to be established then linked to potential physiological mechanisms.

## Effects of age on the expressed dynamics

Our results suggest that age is not a modulating factor of the generating dynamics underlying constant force production and its adaptation to increased force level. Indeed, young and elderly had comparable deterministic (drift coefficients) and stochastic (diffusion coefficients) components across bins. This finding is somewhat surprising since the elderly's performance indicators were clearly different than the young, showing lower accuracy and higher variability. The elderly's force signals had also a different time structure compared to the younger adults' group, with more predictability across time-scales and therefore reflecting a less complex behavior. One must acknowledge however that it is far from trivial how MSE curves relate to the dynamics. Recall, the derivation of the dynamic components is based on the conditional probability (or transition) matrix with a time step of one sample, while MSE seeks for repetitions of a particular motive, and that within a given range *r* (see Methods), and in addition assesses temporal scales beyond a single time step.

It is noticeable that while the diffusion coefficients of elderly were on average larger than those of the young adults, they were also very variable between elderly, which probably prevented the difference in means to reach significance (Fig 5). This inter-individual variability in the elderly was especially observed at the lower (10% MVC) force, and in the force modulation task, wherein age significantly increased the diffusion coefficient, and showed complex interactions for both coefficients with the frequency and force factors that depended as well on the studied bin (i.e., movement phase).

Overall, our results suggest that older and younger adults' force control is governed by the same dynamical mechanisms, but that older adults show an increased stochasticity in the cyclic force modulation task, which echoes with their more variable performance (variance, RMSE) and more complex (MSE) behavioral signals as compared to the younger group. It might be in part due to the slower contractile characteristics of the aging muscles [62], which influences the elderly's ability to ramp-up their produced force. Indeed, compared to younger adults, older adults seem to increase their force less smoothly with multiple bursts, resulting in force outputs of large increments (see for an overview [63]). Thus, for elderly, a sine-wave tracking task imposing relatively small and smooth increments would be more challenging to perform than maintaining a constant force level. From a dynamical perspective, it was argued that elderly are less adaptable than their younger counterparts to the time-dependent demands of continuous force output [64]. To that, one could add the age-related perceptual-motor slowing that reduces visual information processing capacity [44,65], which may have interfered with the elderly's ability to modulate precisely their produced force to match a time-varying target. These physiologic and sensorimotor control changes reduce the ability with aging to use the faster time scales of sensorimotor control. It could explain the observed increased prominence of the stochastic component in older adults in cyclic force modulation tasks, together with the higher behavioral variability reflected by the other conventional measures, as well as the task-

dependent complexity changes. However, as the diffusion coefficient results did not go hand in hand with the other measures in the constant task modality, we cannot conclusive on the presence of a systematic relation between performance, complexity and dynamics-based metrics.

## Conclusion

Our results showed an increase in force signal variability with force level and age, especially so in the modulation task where a regular force modulation was required. This confirms previously reported effects, and extends them to different task conditions [3,18,41,66,67].

By analyzing the multiscale time-structure of force fluctuations we confirmed that, compared to older adults, younger adults have a more complex and adaptable behavior that can be efficiently modulated to meet different types of task constraints. Namely, the young group expressed task-relevant complexity levels, according to which their force signals fluctuated more freely in the constant force conditions, with higher structured irregularity. The reverse was observed in the cyclic force modulation task, wherein behavioral fluctuations of the young participants were much more constrained than the elderly to match the regularity imposed by target, especially so in the low frequency condition. Looking at mean SampEn values, force signals appeared to get less differentiated between task modalities and conditions with aging. Indeed, compared to young participants, elderly showed a reduced modulation of entropy when going from constant to cyclic force tracking (25% less), and no effect of changing the force modulation frequency within the latter task. This finding corroborates previous studies arguing in favor of a declined adaptability of sensorimotor control to task dynamics with aging [18,24,59,64] and gives evidence through various task conditions.

The investigation of the underlying diffusion/stochastic and drift/deterministic components of the dynamics added to this account while tempering age-related declines. It mainly showed comparable generating mechanisms in both age groups (fixed-point for the maintenance task and limit-cycle for the modulation task). Aging had a statistically significant effect only on the force modulation task, and especially on the stochasticity of behavior. This evokes potentially the presence of compensatory mechanisms that preserves the dynamics in less challenging task contexts (i.e., constant force target). We also revealed that both components of the dynamics showed a largely similar pattern and increased with increasing frequency. The stochastic component was the smallest around the local minima of the deterministic component, indicating that the slow(er) dynamics are more stochastic. Furthermore, although we could not unambiguously define the control parameter underlying force control, we could at least exclude force level, and promote force modulation frequency as a potential candidate. Finally, we highlight the increased inter-individual variability amongst the elderly, making it more difficult to extract clear statistically significant effects and unambiguously identify the dynamical mechanisms underlying the observed age-differences in performance and time-structure of fluctuations.

Overall, this work offers a baseline to understand how young and older adults adapt their force control to task requirements. It also shows the complementarity of different methodological and conceptual perspectives in revealing different aspects of sensorimotor adaptation and aging. The better understanding of age-related force control deficits and their manifestation is of great relevance for healthcare interventions in the elderly, as well as the silver economy sector developing different gadgets that must meet the needs and the capabilities of the seniors. It is a well-established practice in geriatrics to test maximal grip force, which here we show that it does not account by itself for age-related declines in force control. With the development of telemedicine and actimetry directed to monitoring patients over longer periods of time with

the use of wearable medical devices, the used paradigm and analysis methods could be integrated in a user-friendly interface to offer a more comprehensive assessment tool for clinicians. It can also be implemented with a gamification effort to train older adults and stimulate their force control capacities. Our study presents some limitations that need to be addressed in future work. Examining for instance a larger cohort of advanced age participants would help in clarifying the effects that did not reach statistical significance, most likely due to the great inter-individual variability of the older group. Additionally, including participants with different levels of physical and cognitive fitness, as well as adding electrophysiological recordings, could help unravel eventual compensation strategies, as well as the moderators of a higher behavioral functioning.

## Acknowledgments

We would like to thank the medical doctors of the research unit in the departmental gerontological center of Marseille (CGD13) for their help with participants' recruitment and assessment, Marine Julien-Vintrou and André Jacques for their technical support, as well as Louise Devillers-Réolon for her help in data collection.

## Author Contributions

**Conceptualization:** Raoul Huys, Jean-Jacques Temprado, Rita Sleimen-Malkoun.

**Data curation:** Hester Knol.

**Formal analysis:** Hester Knol.

**Funding acquisition:** Jean-Jacques Temprado, Rita Sleimen-Malkoun.

**Investigation:** Hester Knol.

**Methodology:** Raoul Huys, Rita Sleimen-Malkoun.

**Project administration:** Jean-Jacques Temprado, Rita Sleimen-Malkoun.

**Resources:** Jean-Jacques Temprado.

**Supervision:** Rita Sleimen-Malkoun.

**Writing – original draft:** Hester Knol, Rita Sleimen-Malkoun.

**Writing – review & editing:** Raoul Huys, Jean-Jacques Temprado, Rita Sleimen-Malkoun.

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
