## [Decision Letter · Decision Letter 0]

10 Sep 2019

PONE-D-19-20040

Performance, complexity and dynamics of force maintenance and modulation in young and older adults

PLOS ONE

Dear Dr. Sleimen-Malkoun,

Thank you for submitting your manuscript to PLOS ONE. After careful consideration, we feel that it has merit but does not fully meet PLOS ONE’s publication criteria as it currently stands. Therefore, we invite you to submit a revised version of the manuscript that addresses the points raised during the review process.

We would appreciate receiving your revised manuscript by Oct 25 2019 11:59PM. To enhance the reproducibility of your results, we recommend that if applicable you deposit your laboratory protocols in protocols.io, where a protocol can be assigned its own identifier (DOI) such that it can be cited independently in the future. For instructions see: http://journals.plos.org/plosone/s/submission-guidelines#loc-laboratory-protocols

We look forward to receiving your revised manuscript.

Kind regards,

Yih-Kuen Jan, PhD

Academic Editor

PLOS ONE

Journal Requirements:

Additional Editor Comments (if provided):

Reviewers' comments:

Reviewer's Responses to Questions

**Comments to the Author**

1. Is the manuscript technically sound, and do the data support the conclusions?

Reviewer #1: Yes

Reviewer #2: Yes

Reviewer #3: Yes

2. Has the statistical analysis been performed appropriately and rigorously? 

Reviewer #1: Yes

Reviewer #2: Yes

Reviewer #3: No

3. Have the authors made all data underlying the findings in their manuscript fully available?

Reviewer #1: Yes

Reviewer #2: Yes

Reviewer #3: No

4. Is the manuscript presented in an intelligible fashion and written in standard English?

Reviewer #1: Yes

Reviewer #2: Yes

Reviewer #3: Yes

5. Review Comments to the Author

Reviewer #1: The topic is interesting. However, there are some minor concerns might need to be clarified.

1. In page 4, the authors mentioned “we aimed for a broad characterization of force control under different task conditions in young and older adults. To do so we combined conventional statistical measures of performance (mean force, variance, coefficient of variation and root mean squared error) with measures of force signals’ complexity (MSE) and dynamics (stochastic and deterministic components). …..”, please describe the purpose more clearly.

2. In page 5, the subjects divided into two group, which are young participants (YN: 18; mean age ± SD = 24.9 ± 3.56 years; 6 females), and elderly participants (EL: 12; mean age ± SD= 76.69 ±6.41 years; 8 females). Do the authors consider the gender effect?

3. Page 5, line 140, the term “LabView” should be “LabVIEW”.

4. How many trails for a subject and if they can familiar the trail before test recording ?

5. The “Results” section include the results of MVC, General characteristics of force performance, Multi Scale Entropy and Drift and diffusion coefficients. If the results can be present in tables, it would be better to realize the results.

6. In page 12, the description “Fig 2. General characteristics of force production. Mean force in the constant task (a) and the modulation task (b). Standard deviation (c) and coefficient of variation (e) for the constant task. Mean relative phase (d) and uniformity (f) for the modulation task. Black bars represent the results of the young group, and white bars those of the elderly. Error bars represent the standard deviation.” Maybe follow the order (a)….(b)….(c)…(d)….(e)….(f)…would be better.

7. In page 13, the description “Fig 3. Root mean square error (RMSE) of produced force. young (white) and elderly (black) participants in the constant”? It is different from that show in Fig. 3 young (black) and elderly (white).

8. In page 14, the description “Fig 4. Multiscale entropy of produced force. Mean sample entropy for constant (a) and modulation (b) tasks, for young (black) and elderly participants (black)” It is different from that show in Fig. 4 young (black) and elderly (white).

9. If the printing version is in black-and-white, the color figures should be revised.

10. Maybe the author can describe the application of their finding in the manuscript.

Reviewer #2: Aging increased the stochasticity (noise-driven fluctuations) of force fluctuations in the cyclic force modulation, which could be related to the increased complexity found in the elderly for this same task. This findings of this study show their complementarity in revealing distinct aspects of sensorimotor adaptation to task constraints and age-related declines. Further research is still needed to identify the physiological underpinnings. The authors may make a paragraph to direct future works.

The authors used two tasks requiring either force maintenance (straight line target force) or force modulation (sine-wave target force) around different force levels and at different modulation frequencies. It is not easy to know the procedure of manipulation activities. The authors may consider to make a photo or sketch to illustrate the procedure of two manipulation activities. Figure 1 is not clear to know the detail.

A height-adjustable force transducer was affixed to the experimental table. The participant could comfortably grasp it while being seated with their arms resting on the table. The authors may make a photo to illustrate how the force transducer affixed to the table, and how the subjects comfortably grasp it. Figure 1 is not clear to know the detail.

The authors may list the analysis formula (i.e., root mean squared error, multiscale entropy) of this study. It would be easy to know how to calculate these parameters.

Reviewer #3: The authors conducted this study to quantify the force control ability of different force production tasks in young and elder people based on different perspectives, including accuracy, variability, complexity, and dynamics. The current findings set up a foundation for further study about motor control ability in people of different ages. However, the limitations of this study are not described so that readers might overinterpret the results. And in my opinion, the clinical application should be clarified to increase the value of the results.

MAJOR COMMENT

There is no need to put any term like “see”, “as in” etc. before the citation number. Therefore, please delete all of them.

Introduction

In general, the introduction section is very comprehensive. It seems like authors focus a lot on introducing the application of entropy-based metric and dynamics on physiological signals, indicating the complexity and dynamic aspects of signals. I suggest that the authors should emphasize more on the importance of applying these analysis methods to the medical signals, and the purpose of this study would be more convincing. Moreover, the aim of this study does not have to be conveyed in the first paragraph. The first paragraph just needs to reveal the signal complexity and dynamics are essential in evaluating human performance.

Results

After reading all statistic results, I am not certain how the authors performed ANOVA. Basically, the main result of ANOVA, which contains 2 or more factors, is if there is a significant interaction between factors first. Later the main effect of each factor could be examined. Last the post-hoc test would be carried out to make multiple comparisons. Take the result of constant force as an example. However, the authors reported there is a significant difference in the mean applied force between force levels first. Subsequently, the ANOVA shows an interaction between force level and age. Furthermore, the figure 2a seems not to have an interaction between age and force level. I am confused about what statistic analysis the authors used to derive the first and second result. Each variable which is reported later in this manuscript also confused me a lot.

Discussions

The discussion section is well written. I consider it might be better that the first two paragraphs of conclusion are summarized in the discussion section. Also, please discuss the limitations of this study.

MINOR COMMENT

Introduction

Line 100: Please add the full name of “NMSS” when first using.

Methods

Line 131-134: How did you define young and elderly participants? And please report body weight and height of all participants.

Line 244: Are t-tests used as post hoc tests? Please specify what test you used for post hoc tests.

Results

Some p values here are reported to be 0.000. The p value is impossible to be 0.000, so please revise all of them into “p<0.001”.

Line 258: “p=.0018”: Based on other p values reported, if the authors want to present the value with three decimal places, please be consistent. “0.0018” should be round to three decimal places

Line 388, Line 403 and Line 407: I guess “bins [3, 8]” indicates bin is 3 and 8. But the square brackets confuse readers because brackets are also used for citation. Please use other symbols or expressions.

Figure 2. Please add the full name of MVC in legend.

Figure 3. Please add the full name of MVC in legend.

Figure 4. Please add the full name of MVC and CMSE in legend.

Figure 5. Please add the full name of MVC in legend.

Figure 6. Please add the full name of MVC in legend.

6. PLOS authors have the option to publish the peer review history of their article (what does this mean?). If published, this will include your full peer review and any attached files.

Reviewer #1: No

Reviewer #2: Yes: Chi-Wen Lung

Reviewer #3: No

---

## [Author Response · Author response to Decision Letter 0]

7 Oct 2019

Our responses can be found in the rebuttal letter uploaded as "Response to reviewers" .pdf file.

---

## [Decision Letter · Decision Letter 1]

30 Oct 2019

PONE-D-19-20040R1

Performance, complexity and dynamics of force maintenance and modulation in young and older adults

PLOS ONE

Dear Dr. Sleimen-Malkoun,

Thank you for submitting your manuscript to PLOS ONE. After careful consideration, we feel that it has merit but does not fully meet PLOS ONE’s publication criteria as it currently stands. Therefore, we invite you to submit a revised version of the manuscript that addresses the points raised during the review process.

We would appreciate receiving your revised manuscript by Dec 14 2019 11:59PM. To enhance the reproducibility of your results, we recommend that if applicable you deposit your laboratory protocols in protocols.io, where a protocol can be assigned its own identifier (DOI) such that it can be cited independently in the future. For instructions see: http://journals.plos.org/plosone/s/submission-guidelines#loc-laboratory-protocols

We look forward to receiving your revised manuscript.

Kind regards,

Yih-Kuen Jan, PhD

Academic Editor

PLOS ONE

Reviewers' comments:

Reviewer's Responses to Questions

**Comments to the Author**

1. If the authors have adequately addressed your comments raised in a previous round of review and you feel that this manuscript is now acceptable for publication, you may indicate that here to bypass the “Comments to the Author” section, enter your conflict of interest statement in the “Confidential to Editor” section, and submit your "Accept" recommendation.

Reviewer #1: All comments have been addressed

Reviewer #2: All comments have been addressed

Reviewer #3: All comments have been addressed

2. Is the manuscript technically sound, and do the data support the conclusions?

Reviewer #1: Yes

Reviewer #2: Yes

Reviewer #3: Yes

3. Has the statistical analysis been performed appropriately and rigorously? 

Reviewer #1: Yes

Reviewer #2: Yes

Reviewer #3: Yes

4. Have the authors made all data underlying the findings in their manuscript fully available?

Reviewer #1: Yes

Reviewer #2: Yes

Reviewer #3: Yes

5. Is the manuscript presented in an intelligible fashion and written in standard English?

Reviewer #1: Yes

Reviewer #2: Yes

Reviewer #3: Yes

6. Review Comments to the Author

Reviewer #1: The authors have been addressed all the comments. It may be a good research article to be published.

Reviewer #2: This is a simple task for young and older adults. Authors may consider making an interesting story or purpose to promote this task for reader reading attractively.

Reviewer #3: Thanks for authors' response. I still have one more question from your revision.

I am not sure your definition of "the corrected t-test" used in this study as the post hoc test. I guess you probably directly used the bonferroni test in SPSS to perform multiple comparisons so that you don't need to report the adjusted p value. But if using t-tests with an adjusted significant level (we call it as bonferroni correction), you must report adjusted p value. Either way is correct. Please specify.

7. PLOS authors have the option to publish the peer review history of their article (what does this mean?). If published, this will include your full peer review and any attached files.

Reviewer #1: No

Reviewer #2: Yes: Chi-Wen Lung

Reviewer #3: No

---

## [Author Response · Author response to Decision Letter 1]

5 Nov 2019

Reviewer #2:

This is a simple task for young and older adults. Authors may consider making an interesting story or purpose to promote this task for reader reading attractively.

Response: Following the reviewer’s suggestion, we now put more forward the functional and clinical relevance of the task, by adding the following to what was already mentioned in the conclusion section.

- End of the abstract: “Although further research is still needed to identify the physiological underpinnings, the used task and methodology is shown to have both fundamental and clinical applications.”

- First paragraph of the introduction: “In the present work, we adopted a broad perspective to characterize force control in young and older adults in a functionally relevant task that can be quantitatively manipulated to scale the imposed attentional and sensorimotor requirements.”

- First paragraph of the discussion: “The used force control task involves multiple control processes (e.g., force production, visuomotor integration, and attention) that are implicated in daily living activities, and presents also the advantage of being easy to implement and undergo for both research and clinical purposes.”

Reviewer #3:

Thanks for authors' response. I still have one more question from your revision.

I am not sure your definition of "the corrected t-test" used in this study as the post hoc test. I guess you probably directly used the bonferroni test in SPSS to perform multiple comparisons so that you don't need to report the adjusted p value. But if using t-tests with an adjusted significant level (we call it as bonferroni correction), you must report adjusted p value. Either way is correct. Please specify.

Response : The reviewer’s guess is correct. We did use the Bonferroni test in SPSS. This is now clearly specified in the methods section.

---

## [Decision Letter · Decision Letter 2]

18 Nov 2019

Performance, complexity and dynamics of force maintenance and modulation in young and older adults

PONE-D-19-20040R2

Dear Dr. Sleimen-Malkoun,

We are pleased to inform you that your manuscript has been judged scientifically suitable for publication and will be formally accepted for publication once it complies with all outstanding technical requirements.

With kind regards,

Yih-Kuen Jan, PhD, University of Illinois at Urbana-Champaign

Academic Editor

PLOS ONE

Additional Editor Comments (optional):

Reviewers' comments:

Reviewer's Responses to Questions

**Comments to the Author**

1. If the authors have adequately addressed your comments raised in a previous round of review and you feel that this manuscript is now acceptable for publication, you may indicate that here to bypass the “Comments to the Author” section, enter your conflict of interest statement in the “Confidential to Editor” section, and submit your "Accept" recommendation.

Reviewer #1: All comments have been addressed

Reviewer #3: All comments have been addressed

2. Is the manuscript technically sound, and do the data support the conclusions?

Reviewer #1: Yes

Reviewer #3: Yes

3. Has the statistical analysis been performed appropriately and rigorously? 

Reviewer #1: Yes

Reviewer #3: Yes

4. Have the authors made all data underlying the findings in their manuscript fully available?

Reviewer #1: Yes

Reviewer #3: Yes

5. Is the manuscript presented in an intelligible fashion and written in standard English?

Reviewer #1: Yes

Reviewer #3: Yes

6. Review Comments to the Author

Reviewer #1: 1.The maunscript includes revised version last time, maybe it should be corrected.

2.Page 37, double reference no. 66, it needs to be modified.

3.If possible, highlight the revised parts would be better.

Reviewer #3: (No Response)

7. PLOS authors have the option to publish the peer review history of their article (what does this mean?). If published, this will include your full peer review and any attached files.

Reviewer #1: No

Reviewer #3: No

---

## [Editor Report · Acceptance letter]

21 Nov 2019

PONE-D-19-20040R2 

Performance, complexity and dynamics of force maintenance and modulation in young and older adults 

Dear Dr. Sleimen-Malkoun:

I am pleased to inform you that your manuscript has been deemed suitable for publication in PLOS ONE. Congratulations! Your manuscript is now with our production department. 

With kind regards,

on behalf of

Dr. Yih-Kuen Jan 

Academic Editor

PLOS ONE